# Alternating current electromagnetic field exposure lessens intramyocellular lipid accumulation due to high-fat feeding via enhanced lipid metabolism in mice

**Ryosuke Nakanishi[1,2], Masayuki Tanaka[1,3], Badur un Nisa[1], Sayaka Shimizu[1], Takumi Hirabayashi[1], Minoru Tanaka[1,4], Noriaki Maeshige[1], Roland R. Roy[5], Hidemi Fujino[1]***

1 Department of Rehabilitation Science, Kobe University Graduate School of Health Sciences, Kobe, Japan, 2 Department of Physical Therapy, Kobe International University, Kobe, Japan, 3 Department of Physical Therapy, Okayama Healthcare Professional University, Okayama, Japan, 4 Department of Rehabilitation Science, Osaka Health Science University, Osaka, Japan, 5 Brain Research Institute and Department of Integrative Biology and Physiology, University of California, Los Angeles, Los Angeles, CA, United States of America

* fujino@phoenix.kobe-u.ac.jp

**Data Availability Statement:** All relevant data are within the paper and its Supporting Information files.

## Abstract

Long-term high-fat feeding results in intramyocellular lipid accumulation, leading to insulin resistance. Intramyocellular lipid accumulation is related to an energy imbalance between excess fat intake and fatty acid consumption. Alternating current electromagnetic field exposure has been shown to enhance mitochondrial metabolism in the liver and sperm. Therefore, we hypothesized that alternating current electromagnetic field exposure would ameliorate high-fat diet-induced intramyocellular lipid accumulation via activation of fatty acid consumption. C57BL/6J mice were either fed a normal diet (ND), a normal diet and exposed to an alternating current electromagnetic field (ND+EMF), a high-fat diet (HFD), or a high-fat diet and exposed to an alternating current electromagnetic field (HFD+EMF). Electromagnetic field exposure was administered 8 hrs/day for 16 weeks using an alternating current electromagnetic field device (max.180 mT, Hokoen, Utatsu, Japan). Tibialis anterior muscles were collected for measurement of intramyocellular lipids, AMPK phosphorylation, FAT/CD-36, and carnitine palmitoyltransferase (CPT)-1b protein expression levels. Intramyocellular lipid levels were lower in the HFD + EMF than in the HFD group. The levels of AMPK phosphorylation, FAT/CD-36, and CPT-1b protein levels were higher in the HFD + EMF than in the HFD group. These results indicate that alternating current electromagnetic field exposure decreases intramyocellular lipid accumulation via increased fat consumption.

## Introduction

Long-term high-fat feeding results in lipid accumulation in non-adipose tissue, such as skeletal muscle (intramyocellular lipids) [1]. Intramyocellular lipid accumulation induces increased

**Funding:** This study was supported by Grants-in-Aid for Scientific Research from the Japanese Ministry of Education, Culture, Sports, Science, and Technology (JSPS KAKENHI, grant number 20K19331, 20KK0227, 22H03453 and 22K19752). Ryosuke Nakanishi (20K19331) is involved in all processes involved in this experiment (conceived, designed, performed, analyzed, interpreted results, prepared figures, drafted manuscript, edited and revised manuscript). Hidemi Fujino (20KK0227, 22H03453 and 22K19752) contributed to conceiving and designing this experiment and correcting the manuscript. In addition, the author supervised all of this experiment as the corresponding author.

**Competing interests:** The authors have declared that no competing interests exist.

insulin resistance and altered glucose metabolism [2, 3]. Insulin resistance and altered glucose metabolism are present in many metabolic disorders, such as type 2 diabetes mellitus and metabolic syndrome [4, 5].

Intramyocellular lipid accumulation is associated with an energy imbalance between excess fat intake and fatty acid consumption [6, 7]. The fatty acid transporter CD-36 (FAT/CD-36) and carnitine palmitoyl-transferase-1 (CPT-1) are essential for transporting fatty acids into the mitochondria and consuming fatty acids [8, 9]. Fatty acid consumption also is regulated by the tricarboxylic acid (TCA) cycle and uncoupling of the mitochondria [9–11]. Citrate synthase (CS) and succinate dehydrogenase (SDH) activities, which are indicators of TCA cycle activity in mitochondria, are important energy consumption processes and increase reducing equivalents [9, 10]. Uncoupling proteins (UCPs) constitute a mitochondrial carrier proteins subgroup (UCP-1–UCP-5) located in the inner mitochondrial membrane [12]. UCPs play a role in thermogenesis by uncoupling mitochondrial respiration, which controls energy consumption [13]. In particular, UCP-3 is involved in the control of energy consumption in skeletal muscle [11]. Previous studies indicate that endurance exercise and muscle contraction using electrical stimulation increase these factors, thus increasing fatty oxidation consumption [14–16]. Furthermore, exercise prevents intramyocellular lipid accumulation, improves glycemic control, reduces body fat, and increases insulin sensitivity [17, 18]. On the other hand, severe peripheral neuropathy caused by long-term high-fat feeding likely increases the risk of skin ulceration with exercise, including weight-bearing [17]. Therefore, alternate intervention strategies are necessary to stimulate muscle fat oxidation enzymes and attenuate the accumulation of intramyocellular lipids.

Alternating current electromagnetic field exposure results in a variety of biological effects. For example, electromagnetic field exposure accelerates electron transport from cytochrome C to cytochrome oxidase in rat livers [19]. The electromagnetic field also increases mitochondria metabolism in human sperm [20]. Increasing mitochondrial energy metabolism plays a central role in fatty acid consumption, leading to less intramyocellular lipid accumulation [21]. Therefore, an electromagnetic field may increase fatty acid consumption following long-term high-fat feeding without the need for muscle contraction and/or weight-loading exercise. The present study was designed to determine whether an alternating current electromagnetic field could effectively prevent intramyocellular lipid accumulation induced by high-fat feeding via increased FAT/CD-36, CPT-1, and UCP-3 protein expression and SDH and CS activities. Thus, the results of this study may lead to the development of a new countermeasure for high-fat feeding-induced intramyocellular lipid accumulation.

## Materials and methods

### Ethical approval

This study was approved by the Institutional Animal Care and Use Committee and was conducted in accordance with Kobe University's Animal Care and Use Protocol (P180802). The experiments were performed according to the National Institutes of Health Guidelines for the care and use of laboratory animals.

### Experiment protocol

The study was designed to determine (i) whether alternating current electromagnetic field exposure could increase lipid metabolism to attenuate intramyocellular lipid accumulation in the tibialis anterior muscle (TA) induced by long-term high-fat feeding, and (ii) whether alternating current electromagnetic field exposure induced mitochondrial uncoupling in the muscle cells.

## Experiment 1

**Animals and diets.** Male C57/BL6 mice (3 weeks old; Japan SLC, Shizuoka, Japan) were housed in cages in a temperature-controlled room (22 ± 2˚C) with a 12-hour light/dark cycle. The mice were allowed to acclimate to the laboratory environment with free access to a normal diet (CE-2, CLEA Japan, Tokyo, Japan) and water for one week. Mice were divided randomly into four groups: 1) mice fed a normal diet with no alternating current electromagnetic field exposure (ND: 19.4 ± 1.3 g, n = 5), 2) mice fed a normal diet with exposure to an alternating current electromagnetic field (ND+EMF: 19.8 ± 0.4 g, n = 5), 3) mice fed a high-fat diet (HFD, D-12492, Research Diets, New Brunswick, NJ) with no alternating current electromagnetic field exposure (HFD: 18.2 ± 0.4 g, n = 5), or 4) mice fed a high-fat diet with exposure to an alternating current electromagnetic field (HFD+EMF: 19.2 ± 0.4 g, n = 5). The HFD groups had free access to the high-fat diet for 16 weeks, and food intake in the normal diet groups was controlled to match the HFD groups daily. At the end of the experimental period, the rats were anesthetized deeply by inhalation of 4% isoflurane, then the abdominal cavity was opened, and an approximately 2ml blood sample was obtained from the inferior vena cava with a syringe. Blood samples were centrifuged at 3,000 g for 10 min at 4˚C. After blood collection, the mice were killed by intraperitoneal administration of sodium pentobarbital (100 mg/kg). The tibialis anterior muscles and epididymal adipose tissues were removed quickly bilaterally, and the average muscle wet weights were recorded. The tibialis anterior muscles then were frozen immediately in a dry ice acetone bath and stored at −80˚C.

**Alternating current electromagnetic field protocol.** Mice were exposed to an alternating current electromagnetic field from the bottom of the cage throughout the experimental period using an alternating current electromagnetic field stimulator (Hokoen, Utatsu, Japan) (Fig 1A). The alternating current electromagnetic field exposure applied for 8 hrs (during the dark period) per day at a frequency of 60 Hz and maximal amplitude of 180 mT. Animals showed no signs of discomfort, and the temperature inside the cage did not change in response to the electromagnetic field.

**Insulin tolerance tests and oral glucose tolerance tests.** The insulin resistance test (ITT) was conducted on week 14. Food was removed for 5 hrs before insulin (0.5 U/kg BW Humulin; Eli Lilly, Kobe, Japan) was injected intraperitoneally [22]. Whole blood samples (2–3 μL each) were collected from a tail-clip bleed, and glucose was measured by a glucometer (Glutest Neo, Sanwa Kagaku Kenkyuusyo, Nagoya, Japan) at 0, 15, 30, and 45 min after the insulin injection. The oral glucose tolerance test (OGTT) was performed at week 15. After 16 hrs of fasting, glucose was administered orally by a catheter (2.0 g/kg BW) [23]. Whole blood samples (2–3 μL each) were collected from a tail-clip bleed at 0, 30, 60, 90, 120, and 180 min after glucose administration, and the glucose levels were measured at each time point.

**Plasma biochemistry.** The glucose concentrations in the plasma were measured using the BCG method and biuret test (Glucose C-II Test Wako; FUJIFILM Wako Chemicals, Tokyo, Japan) [24]. The cholesterol concentrations were measured using the GPO-DAOS method (Cholesterol E-Test Wako; FUJIFILM Wako Chemicals) [24]. Non-esterified fatty acid (NEFA) levels were measured using the ACS-ACOD method (NEFA C-Test Wako; FUJIFILM Wako Chemicals) [25].

**Histology and immunohistochemical analyses.** Transverse tissue sections (12 μm thick) were cut from the muscle mid-portion in a cryostat (CM-1510S, Leica Microsystems, Mannheim, Germany) at −25˚C and mounted on glass slides. The sections were stained for Oil Red O and succinate dehydrogenase (SDH). The sections were captured with a microscope (CX41; Olympus, Tokyo, Japan, objective lens: x20), and were quantified using Image J software (NIH, Bethesda, MD). Raw images were exported as 600 dpi TIFF files using software after

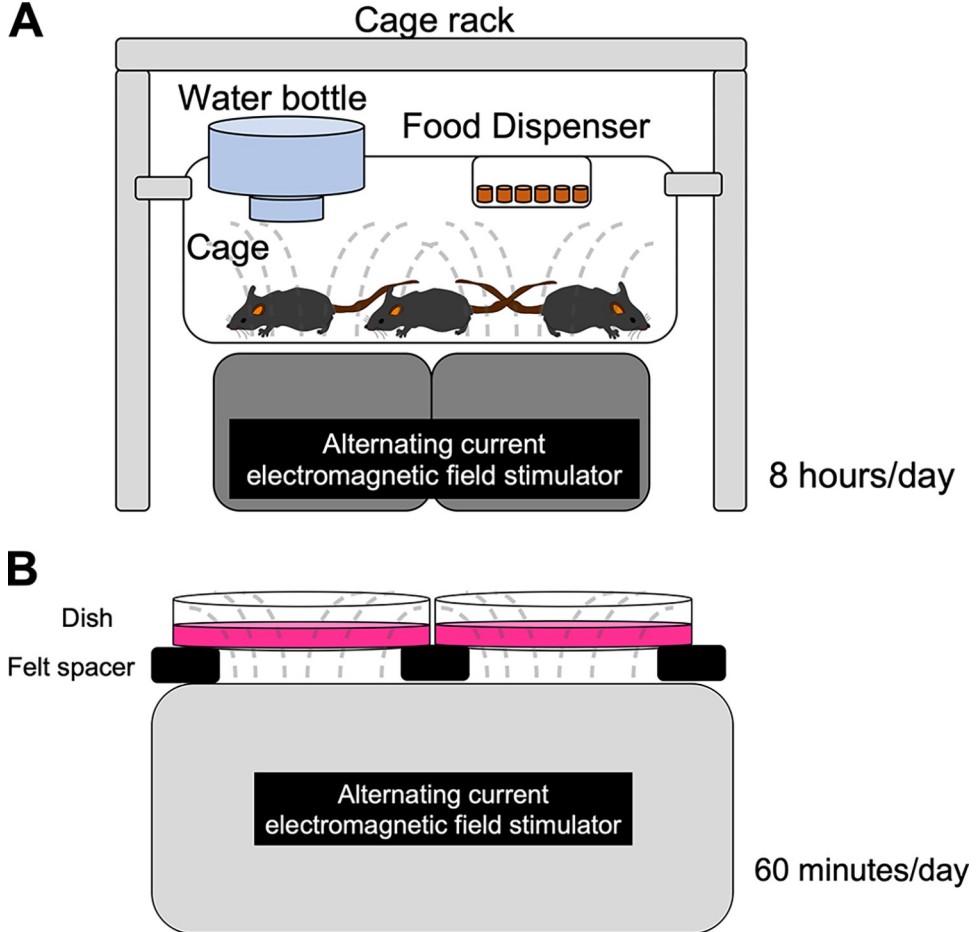

**Fig 1. Illustration of the alternating current electromagnetic field exposure.** An alternating current electromagnetic field was exposed from the bottom of the cages housing the mice for 8 hrs per day (during the dark period). The dotted line represents the image of alternating current electromagnetic field exposure (A). An alternating current electromagnetic field was exposed from the bottom of the dish containing C2C12 myotubes once for 60 min. The dotted line represents the image of alternating current electromagnetic field exposure (B).

quantified. Oil Red O staining was used to detect intramyocellular lipids as previously described [26]. Briefly, serial sections were fixed with 4% paraformaldehyde for 60 min and exposed to an oil Red O working solution (FUJIFILM Wako Chemicals) for 30 min [27]. The amount of intramyocellular lipids in each fiber was calculated randomly within the section for each of the five different fields (92.5mm$^2$/ field) and was quantified as the percentage of the area occupied by Oil Red O-stained droplets (total area occupied by lipid droplets of a muscle fiber × 100/total cross-sectional area of the fiber). SDH activity was analyzed as previously described [25]. Briefly, the sections were incubated in 0.1 M phosphate buffer (pH 7.6) containing 0.9 mM NaN3, 0.9 mM 1-methoxyphenazine methylsulfate, 1.5 mM nitroblue tetrazolium, 5.6 mM EDTA–disodium salt, and 48 mM succinate disodium salt for 45 min at 37°C. The SDH activity was captured randomly within the section for each of the five different fields (92.5mm$^2$/ field), and was converted to 8-bit grayscale and was quantified as a mean gray value.

Immunohistochemical analyses were conducted as previously described [28]. The sections were captured with a fluorescence microscope (BX51; Olympus) and quantified by Image J

software. Raw images were exported as 600 dpi TIFF files using the software after quantified. Briefly, muscle samples were fixed with 4% paraformaldehyde for 30 min, and blocked with 3% bovine serum albumin for 60 min. The sections were exposed overnight at 4°C to FAT/ CD-36 polyclonal antibody (1:100, NB-400-144SS, RRID AB_920879; NOVUS Biologicals, Littleton, CO). The sections then were incubated with a DyLight 488-coupled secondary antibody (1:1000 dilution; Jackson ImmunoResearch, West Grove, PA) and blue fluorescence 4',6-diamidino-2-phenylindole (DAPI; Thermo Fisher Scientific, Waltham MA) for 60 min in the dark room. The cell surface immunofluorescence was captured randomly within the section for each of the five different fields (211.6mm2/ field), and a mean percentage was calculated for each muscle (total area occupied by immunofluorescence of a muscle fiber × 100/total cross-sectional area of the fiber) [29].

**Citrate Synthase (CS) activity.** CS activity was analyzed as previously described [25]. Briefly, the supernatants were solubilized in a reaction buffer containing 0.1 mM 5,5-dithio-bis-(2-nitrobenzoic acid) and 0.3 mM acetyl-CoA. The reaction was initiated by incubating with oxaloacetic acid (0.5 mM final concentration). The absorbance was measured at 412 nm for 5 min.

**Western blot analysis.** Western blot analysis was conducted as previously described [28]. Briefly, tissue samples were homogenized in ice-cold homogenizing buffer (Ez RIPA Lysis kit, WSE-7420, ATTO, Tokyo, Japan). The homogenates were centrifuged at 15,000 g for 30 min at 4°C, solubilized in loading buffer (EzApply, AE-1430, ATTO), and boiled for 10 min at 80°C. Proteins (30 μg/lane) were separated by 10% sodium dodecyl sulfate-polyacrylamide gel electrophoresis (e-PAGEL, ATTO) and then transferred to polyvinylidene fluoride membranes. The membranes were blocked for 60 min in 3% bovine serum albumin in Tris-buffered saline with Tween-20 and then incubated with CD-36 (1:1000 dilution; # ab133625, RRID AB_2716564; Abcam, Cambridge, England), UCP-3(1:1000; #10750-1-AP, RRID AB_2272729; Protein tech, Rosemont, IL), p-AMPKα (Thr172)(1:1000; #2535, RRID AB_2799368; Cell Signaling Technology, Danvers, MA), t-AMPKα (1:1000; #2532, RRID AB_330331; Cell Signaling Technology), or CPT-1b (1:1000; #22170-1-AP, RRID AB_2713959; Protein tech) antibodies overnight at 4°C. The membranes were incubated for 60 min at room temperature with anti-mouse or anti-rabbit IgG antibodies conjugated to horseradish peroxidase (1:5000; HAF018/HAF008, RRID AB_573130/ AB_357235; R&D systems, Minneapolis, MN). Antibody binding was detected using a chemiluminescent reagent (Ez West Lumi One, WSE-7110, ATTO) and analyzed with an image reader (Lumino Graph l; ATTO). Ponceau-S (Beacle, Kyoto, Japan) was used as an internal control.

## Experiment 2

**Cell culture.** Undifferentiated C2C12 cells (myoblasts; American Type Culture Collection, Manassas, VA) were maintained in growth medium (GM) composed of Dulbecco's modified Eagle's medium, 20% fetal bovine serum, 4 mM l-glutamine, and antibiotics (100 IU/ml penicillin, 100 μg/ml streptomycin). The differentiated phenotype (myotubes) was induced by culturing cells in GM supplemented with 2% heat-inactivated horse serum (differentiation medium) for 7–10 days. Both undifferentiated and differentiated C2C12 cell phenotypes were maintained at 37°C in a 5% $CO_2$ humidified atmosphere.

**Alternating current electromagnetic field exposure and sample collection.** Cells were exposed to alternating current electromagnetic field through the bottom of the dish (Fig 1B). The stimulation consisted of exposure to an alternating current electromagnetic field for 60 min at a frequency of 60 Hz and a maximal amplitude of 180 mT. The temperature inside the dish did not change when exposed to the alternating current electromagnetic field.

**Determination of mitochondrial membrane potential.** Mitochondrial membrane potential was determined as previously described [30]. The fluorescence intensities for both monomeric JC-1 forms and J-aggregates were measured at a constant temperature of 25°C with a multifunctional microplate reader (Varioskan LUX, Thermo Fisher Scientific; J-aggregates, excitation/emission 535/595 nm; JC-1 monomers, excitation/emission 485/535 nm; bandwidth ± 12 nm). Fluorescence was recorded at eight locations in each well of 12-well plates (63% of the total area/well). The values of fluorescence acquired from each sample were expressed as the mean ± SD of J-aggregates/JC-1 monomers.

**Real-time PCR analysis.** Total RNA was extracted from C2C12 myotubes using Trizol reagent (Life Technologies, Carlsbad, CA) as previously described [31]. Reverse transcription was carried out using the High-Capacity cDNA Archive Kit (Thermo Fisher Scientific). The cDNAs were used for subsequent quantitative real-time PCR analysis using the SYBR Premix Ex Taq II (Takara Bio, Otsu, Japan). The specific primer sequences were as follows: UCP3: TCAAGCCATGATACGCCTGG (forward) and TGTGATGTTGGGCCAAGTCC (reverse) [32], β-2 microglobulin: CTTTCTGGTGCTTGTCTCACTGA (forward) and GTATGTTCGGCTTCC CATTCTC (reverse) [33]. The PCR reactions were carried out in 48-well microtiter plates on a real-time PCR apparatus (Step One; Thermo Fisher Scientific). The PCR consisted of 95°C (3 mins), 40 cycles at 95°C (10 sec), and 60°C (30 sec) [34]. All specific quantities were normalized to β-2 microglobulin. Data were analyzed using the delta/delta CT method.

## Statistical analyses

Data are reported as means ± SD. The normality of the data was determined using the Kolmogorov-Smirnov test. For Experiment 1 data, a two-way analysis of variance (groups: Normal diets and HFDs × Normal condition and alternating current electromagnetic fields) was used for overall group comparisons. When a group × diet interaction was found, the Tukey test was used to determine any significant differences among the groups. The sample size required for a two-way ANOVA for this study was calculated using G*power 3.1 software (Heinrich Heine University, Dusseldorf, Germany), based on data from a previous study [35]. A total of more than 20 mice were required for this study (effect size = 0.70, α error = 0.05, and power = 0.80). The data for Experiment 2 were analyzed using unpaired two-tailed Student's $t$-tests. For all data, $P$-values less than 0.05 were considered statistically significant. All statistical analyses were performed using GraphPad PRISM software version 7.0 (Intuitive Software for Science, San Diego, CA).

## Results

### Physiological parameters

Daily caloric intake, body weight, and fat mass were higher in both HFD (HFD and HFD +EMF) groups than in both ND (ND and ND+EMF) groups (Table 1). The relative (ratio of muscle to body mass) tibialis anterior muscle masses were lower in both HFD groups than in both ND groups. Exposure to the electromagnetic field had no significant effect on any of these parameters. Fasting blood glucose, cholesterol, and NEFA levels were higher in both HFD groups than in both ND groups and lower in the HFD+EMF group than in the HFD group.

### Oral glucose tolerance test and insulin tolerance test

Blood glucose levels during the OGTT for each group are shown in Fig 2A, and blood glucose levels during the ITT for each group are shown in Fig 2C. The areas under the curves (AUC) for the OGTT and ITT were higher in both HFD groups than in both ND groups. The AUCs for the OGTT and ITT were lower in the HFD+EMF group than in the HFD group (Fig 2B and 2D).

**Table 1. Calorie intake, body mass, fat mass, tibialis anterior absolute and relative muscle mass, and fasting blood glucose, cholesterol, and non-esterified fatty acid levels.**

| | Normal diet | | High-fat diet | |
|---|---|---|---|---|
| | **Non EMF** | **EMF** | **Non EMF** | **EMF** |
| Calorie intake (Kcal/day) | 8.9 ± 1.5 | 8.9 ± 1.5 | 12.7 ± 2.4* | 12.3 ± 2.3* |
| Body mass (g) | 24.2 ± 0.5 | 24.3 ± 1.0 | 39.3 ± 4.1* | 36.5 ± 1.5* |
| Fat mass (mg) | 356.8 ± 15.9 | 402.0 ± 47.5 | 2907.0 ± 415.0* | 2380.8 ± 293.9* |
| Absolute tibial anterior muscle mass (mg) | 42.3 ± 2.8 | 44.0 ± 2.0 | 48.2 ± 1.0 | 44.6 ± 1.9 |
| Relative tibial anterior muscle mass (mg/g) | 1.8 ± 0.2 | 1.8 ± 0.1 | 1.2 ± 0.1* | 1.2 ± 0.1* |
| Fasting blood glucose (mg/dl) | 144.9 ± 13.1 | 152.4 ± 10.2 | 204.4 ± 23.1* | 176.3 ± 7.2*, ♯ |
| Cholesterol(mg/dl) | 130.0 ± 6.8 | 123.5 ± 7.9 | 232.9 ± 16.0* | 196.1 ± 12.9*, ♯ |
| Non-esterified fatty acids (mEq/L) | 0.49 ± 0.08 | 0.47 ± 0.06 | 0.81 ± 0.12* | 0.57 ± 0.06*, ♯ |

Values are expressed as mean ± SD for each group.

* $P < 0.05$ vs. control animals fed the same diet.

# $P < 0.05$ vs. animals fed the normal diet.

## Intramyocellular lipid accumulation in the tibialis anterior muscles

Representative Oil Red O staining patterns in the tibialis anterior muscles from each group are shown in Fig 3A–3D. The percentage of the muscle area showing staining by Oil Red O was greater in both HFD groups than in both ND groups and less in the HFD+EMF group than in the HFD group (Fig 3E).

## Mitochondrial enzyme activity in the tibialis anterior muscles

Representative SDH staining patterns for each group are shown in Fig 4A–4D. The SDH and CS activities were higher in both HFD groups than in both ND groups and higher in the HFD +EMF group than in the HFD group (Fig 4E and 4F).

## Translocation and protein expression of FAT/CD-36 in the tibialis anterior muscles

Representative immunohistochemical staining patterns for FAT/CD-36 in each group are shown in Fig 5A–5L. FAT/CD-36 translocation levels in the tibialis anterior muscles were higher in both HFD groups than in both ND groups. FAT/CD-36 translocation was higher in the HFD+EMF group than in the HFD group (Fig 5M). FAT/CD-36 protein levels were lower in the HFD group than in both ND groups and higher in the HFD+EMF group than in the HFD group (Fig 5N and 5O).

## Levels of phosphorylated AMPK, UCP-3, and CPT-1b in the tibial anterior muscles

Representative Western blots for phosphorylated AMPK, AMPK, CPT-1b, and UCP-3 in each group are shown in Fig 6A. The ratio of phosphorylated AMPK to total AMPK (AMPK phosphorylation) protein level was lower in the HFD group than in both ND groups and higher in the HFD+EMF group than in the HFD group (Fig 6B). CPT-1b and UCP-3 protein levels were higher in the HFD groups than in the ND groups and higher in the HFD+EMF group than in the HFD group (Fig 6C and 6D).

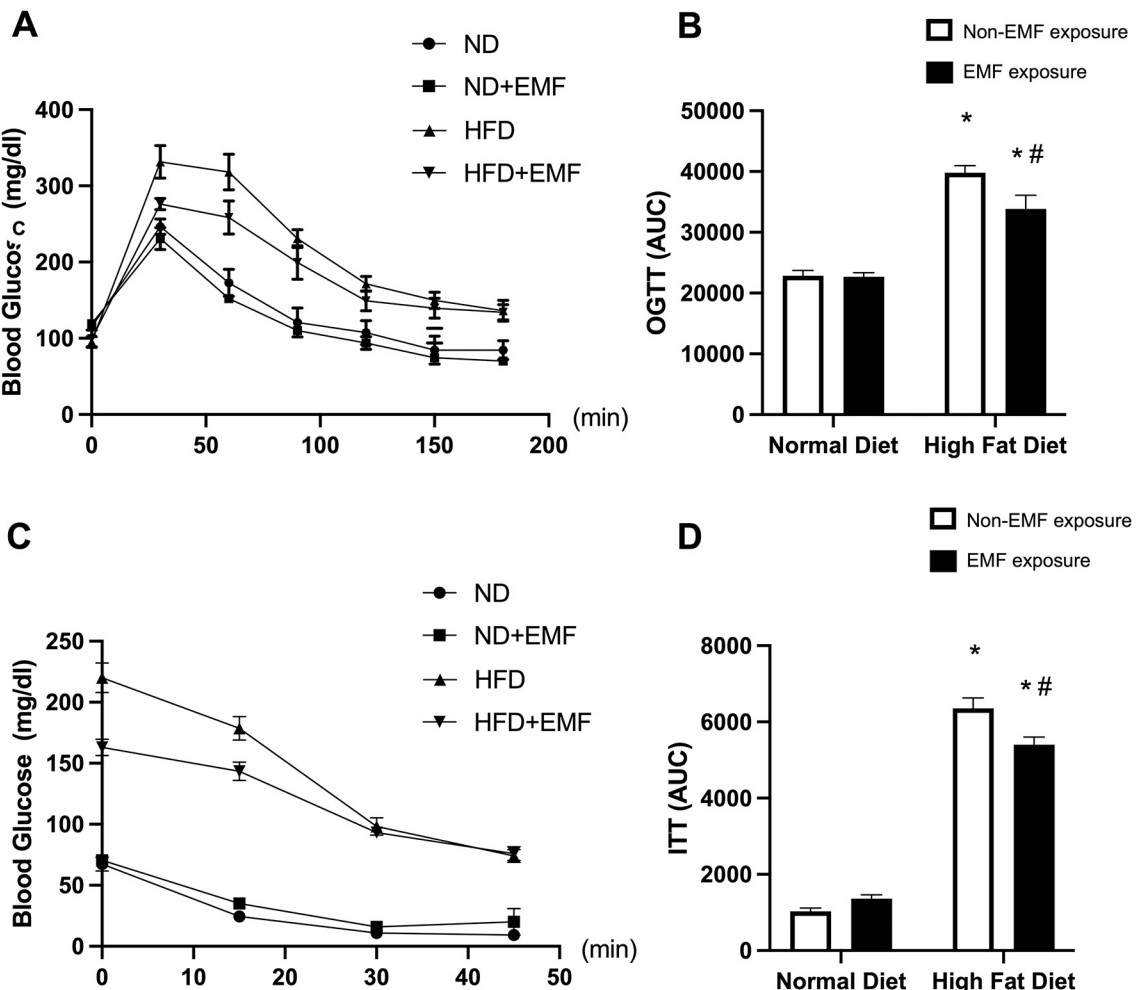

**Fig 2. Oral glucose tolerance test and insulin tolerance test.** Average values for the oral glucose tolerance tests (OGTT) (A) and insulin tolerance tests (ITT) (C) for the ND, ND+EMF, HFD, and HFD+EMF groups. Quantification of the area under the curve (AUC) for the OGTT (B) and ITT (D). Values are expressed as mean ± SD. * *P < 0.05* vs. control animals fed the same diet. # *P < 0.05* vs. animals fed the normal diet. Abbreviations: ND, mice fed a normal diet with no exposure; ND+EMF, mice fed a normal diet with exposure to an alternating current electromagnetic field; HFD, mice fed a high-fat diet with no exposure; HF+EMF, mice fed a high-fat diet with exposure to an alternating current electromagnetic field.

### Mitochondrial membrane potential and UCP-3 mRNA levels in C2C12 myotubes

Representative JC-1 staining images for each group are shown in Fig 7A–7F. The fluorescence intensity for JC-1 in the C2C12 myotubes was lower in the EMF group than in the Non-EMF group (Fig 7G). UCP-3 mRNA levels were higher in the EMF group than in the Non-EMF group (Fig 7H).

## Discussion

The main findings of this study demonstrate that daily exposure to an electromagnetic field lessens the effects of high-fat feeding on fat metabolism in skeletal muscles of adult mice. In contrast, none of these effects were observed in mice fed a normal diet.

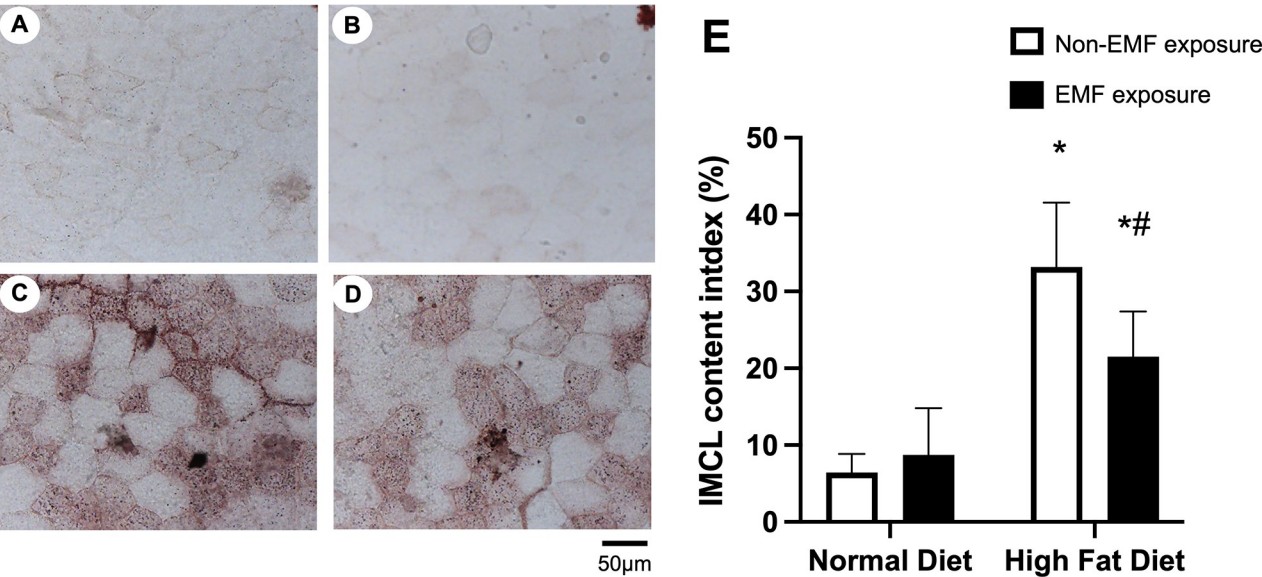

**Fig 3. Intramyocellular lipid accumulation in the tibialis anterior muscles.** Representative images of Oil Red O-stained cross-sections of the tibialis anterior muscles for the ND (A), ND+EMF (B), HFD (C), and HFD+EMF groups (D). The histograms (E) show quantification as a percentage of the area occupied by Oil Red O-stained droplets (total area occupied by lipid droplets of a muscle fiber × 100/total cross-sectional area of the fiber). Values are expressed as mean ± SD. Abbreviations are the same as in Fig 2. * $P < 0.05$ vs. control animals fed the same diet. # $P < 0.05$ vs. animals fed the normal diet.

The high-fat diet resulted in an increase in intramyocellular lipid accumulation, translocation of FAT/CD-36, CPT-1 protein, and SDH and CS levels. Intramyocellular lipid accumulation is associated with an energy imbalance between the intake of excess fat and the consumption of fatty acids [6, 7]. The translocation of FAT/CD-36 and the expression of CPT-1 proteins are required for the oxidation process in skeletal muscle fatty acid consumption and are responsive to energy intake [9, 36–38]. SDH and CS activities are important energy consumption processes that increase reducing equivalents [9, 10]. Messa et al. [1] demonstrated that 16 weeks of high-fat feeding increased the SDH levels in the soleus and extensor digitorum longus muscles of mice. Furthermore, Tuner et al. [39] demonstrated that 20 weeks of high-fat feeding increased the CS levels in the quadriceps muscles of mice. Thus, increasing these factors most likely prevents intracellular lipid toxicity via an adaptive response to the oversupply of fatty acids [6, 7]. However, the accumulation of intramyocellular lipids, insulin resistance, and blood glucose levels increased in the high-fat diet group, despite an increased activation of fatty oxidation and mitochondria metabolism. These results suggest that the oversupply of fatty acids from the high-fat diet exceeded fatty acid consumption, resulting in insulin resistance and hyperglycemia.

Alternating current electromagnetic field exposure inhibited intramyocellular lipid accumulation, insulin resistance, and hyperglycemia under high-fat diet feeding conditions. Intramyocellular lipid accumulation is affected by factors related to energy consumption, such as the transport of fatty acids into the mitochondria, mitochondrial metabolism, and mitochondrial uncoupling [40, 41]. These factors have been shown to interact with and decrease phosphorylated AMP-activated protein kinase (AMPK)α in the muscles of mice in response to long-term feeding with a high-fat diet, consistent with the present results [42–48]. Alternating current electromagnetic field exposure attenuated the decrease in AMPKα phosphorylation and increased FAT/CD-36 translocation and protein expression, CPT-1b protein expression, and SDH and CS activities in mice fed a high-fat diet. AMPKα phosphorylation increases

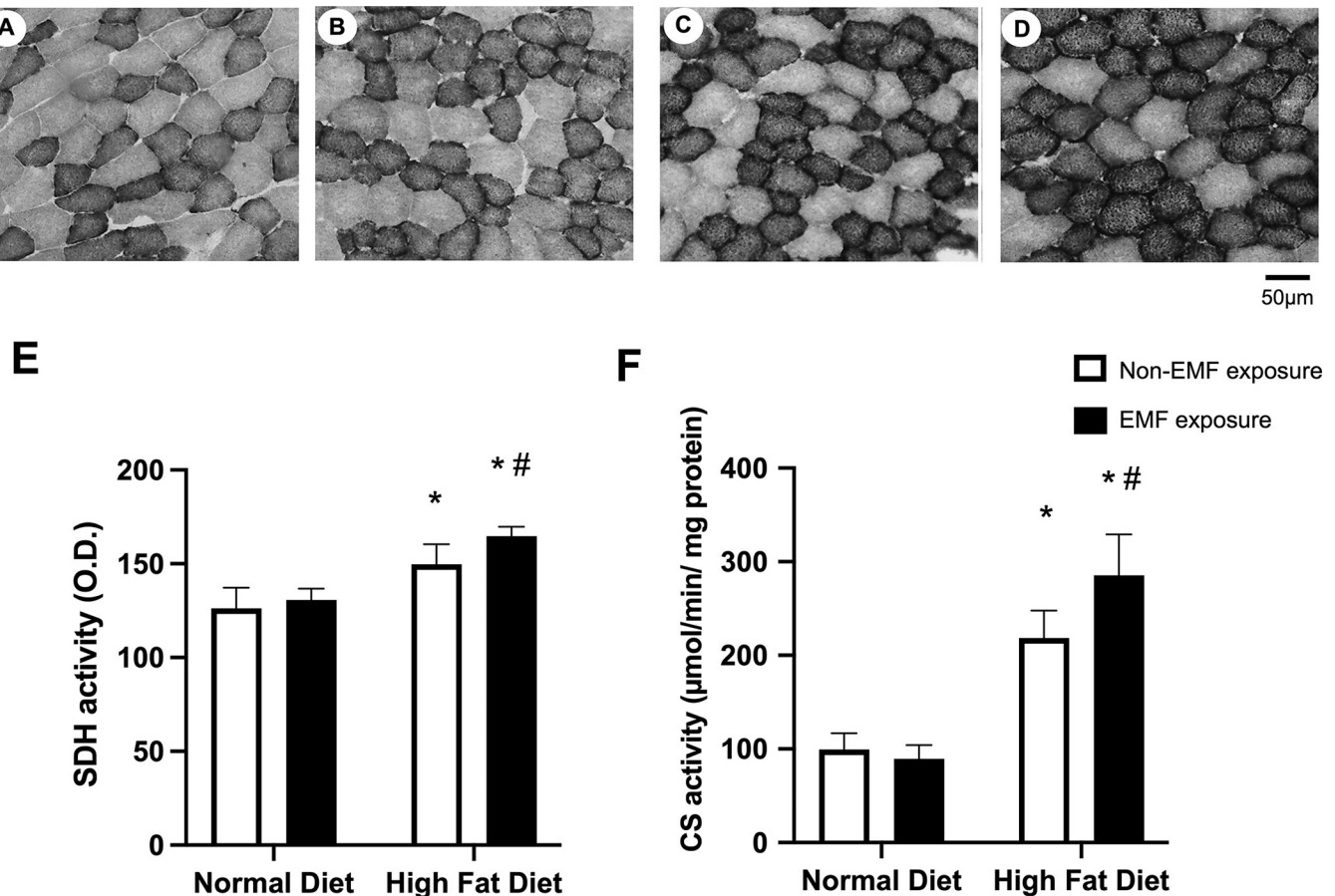

**Fig 4. Mitochondrial enzyme activity in the tibialis anterior muscles.** Representative images of SDH-stained cross-sections of the tibialis anterior muscles for the ND (A), ND+EMF (B), HFD (C), and HFD+EMF groups (D). The histograms for SDH activity show quantification of the staining concentration (E). The histograms for CS activity show quantification of activity intensity (F). Values are expressed as mean ± SD. Abbreviations are the same as in Fig 2. * $P < 0.05$ vs. control animals fed the same diet. # $P < 0.05$ vs. animals fed the normal diet.

FAT/CD-36, which transports fatty acids to the cytosol, and induces translocation to the sarcolemma [42, 43]. AMPK is a heterotrimeric serine-threonine kinase composed of the catalytic α- and noncatalytic β- and γ-subunits [49]. AMPK plays a role in maintaining energy homeostasis and is a target for treating various metabolic disorders [50]. Thus, AMPK activation likely to mitigates metabolic impairments by controlling energy homeostasis. AMPK phosphorylation in C2C12 myotubes increases the mRNA levels of CPT-1, a rate-limiting enzyme that transports fatty acids from the cytosol to the mitochondria [44]. Furthermore, 5-aminoimidazole 4-carboxamide-1-D-ribofuranoside (AICAR), which induces AMPK, increased SDH and CS activities in rat muscles [45, 46]. Therefore, alternating current electromagnetic field exposure may increase the transport of fatty acids into the mitochondria and increase mitochondrial metabolism by inhibiting the decrease in AMPKα phosphorylation induced by a high-fat diet.

AMPKα phosphorylation also upregulates UCP-3 protein expression. Suwa et al. [45] reported that injection of AICAR, an AMPKα inducer, increases UCP-3 protein expression in rat muscles. UCP3 plays a role in increasing fatty acid metabolism and mitochondrial uncoupling in skeletal muscle [14, 15, 51, 52]. Indeed, UCP-3 transgenic mice exhibit lower fat mass, increased fat utilization, and lower mitochondrial membrane potential, which implies

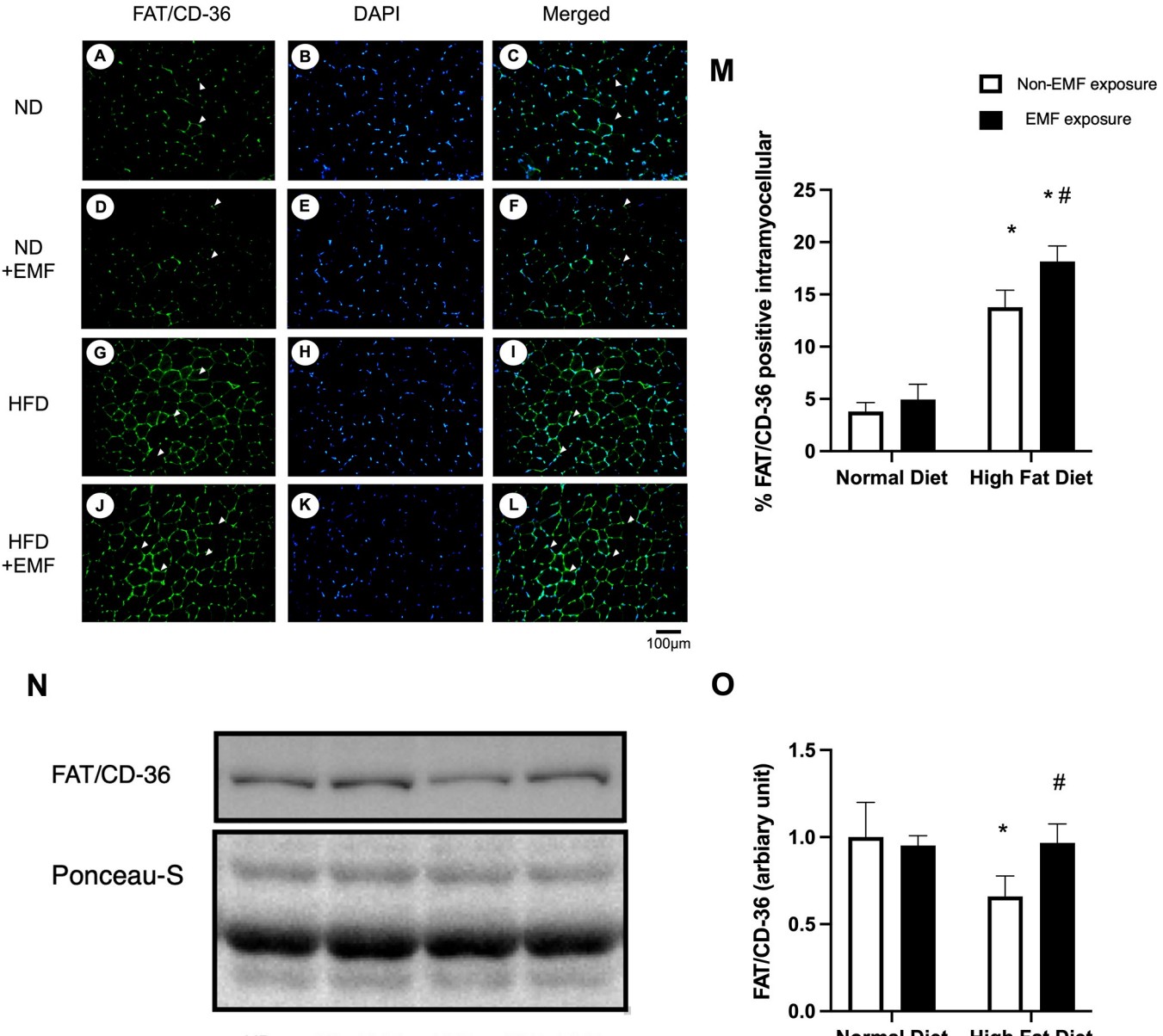

**Fig 5. Translocation and protein expression of FAT/CD-36 in the tibialis anterior muscles.** Representative images of immunohistochemically stained cross-sections of the tibialis anterior muscles for the ND (A, B, and C), ND+EMF (D, E, and F), HFD (G, H, and I), and HFD+EMF groups (J, K, and L). Serial images show the expression of FAT/CD36 (green) (A, D, G, and J), DAPI (blue) (B, E, H, and K), and for the three images merged (C, I, F, and L). The white arrowheads indicate FAT/CD36 positive sarcolemma. FAT/CD36 protein in the tibialis anterior muscles of each group is shown (N). The histograms show quantification of the intramyocellular FAT/CD-36 positive staining (M) and the band densities (O). Values are expressed as mean ± SD. Abbreviations are the same as in Fig 2. * *P < 0.05* vs. control animals fed the same diet. # *P < 0.05* vs. animals fed the normal diet.

mitochondrial uncoupling [53, 54]. Mitochondrial uncoupling increases energy consumption via proton leak, which accounts for a large proportion of resting oxygen consumption in muscles [55]. Tiraby et al. [56] showed that overexpression of UCP-3 resulted in mitochondrial oxygen consumption in the gastrocnemius muscles of mice fed a high-fat diet. In the present study, alternating current electromagnetic field exposure increased UCP-3 protein expression in the tibialis anterior muscle. In addition, alternating current electromagnetic field exposure

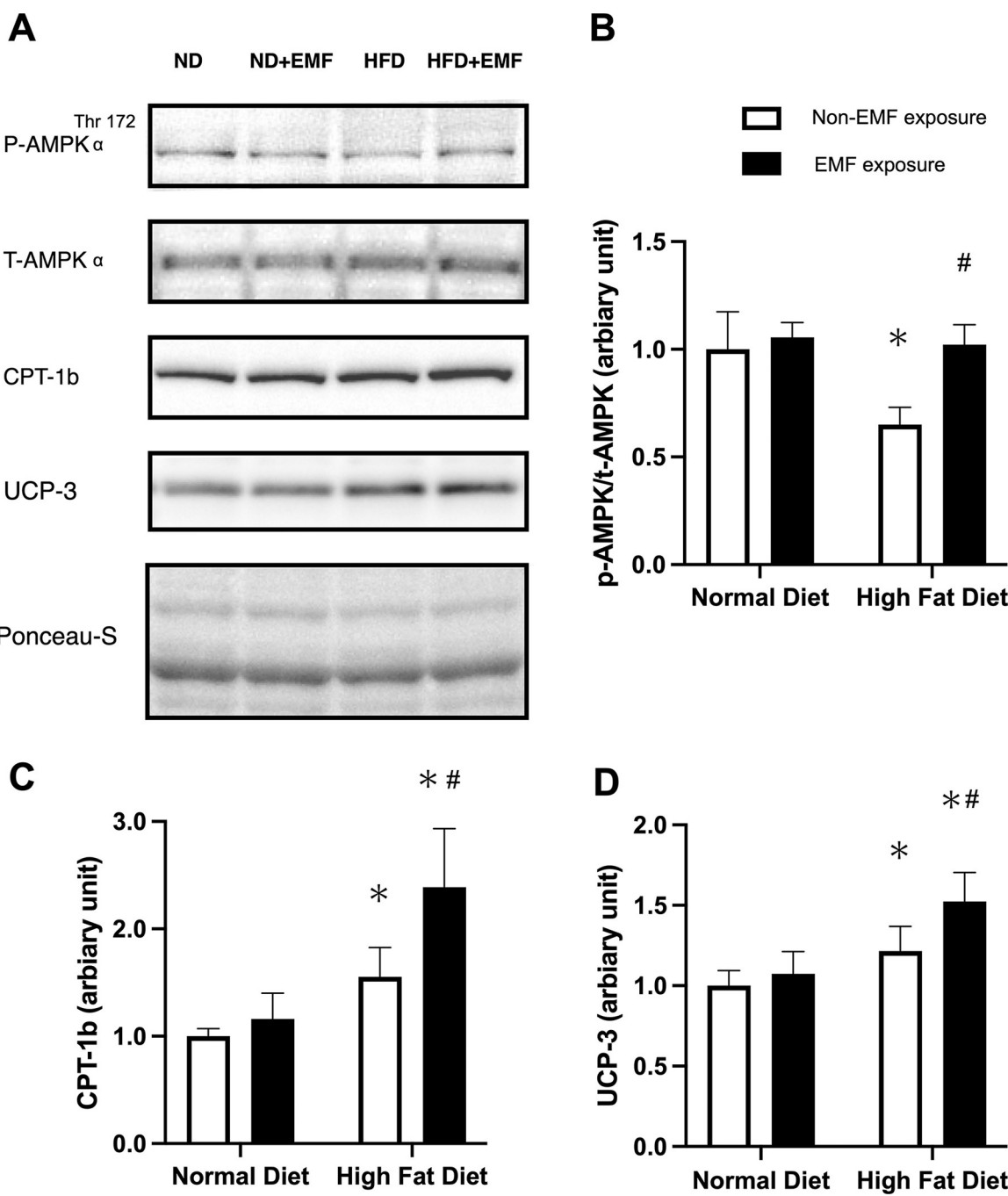

**Fig 6. Levels of phosphorylated AMPK, UCP-3, and CPT-1b in the tibial anterior muscles.** The bands are arranged in order from top to bottom: phospho-AMPK, AMPK, CPT-1, and UCP-3 (A). Phospho-AMPK/AMPK (B), CPT-1b (C), and UCP-3 (D) protein expression levels in the tibialis anterior muscles are shown. Band images are representative of western blots, and the histograms show the quantification of the band densities (D). Values are expressed as mean ± SD. Abbreviations are the same as in Fig 2. * $P < 0.05$ vs. control animals fed the same diet. # $P < 0.05$ vs. animals fed the normal diet.

increased UCP-3 mRNA expression and decreased the membrane potential in C2C12 myotubes. These results suggest that the increase in AMPKα phosphorylation induced by alternating current electromagnetic field exposure resulted in mitochondrial uncoupling via an

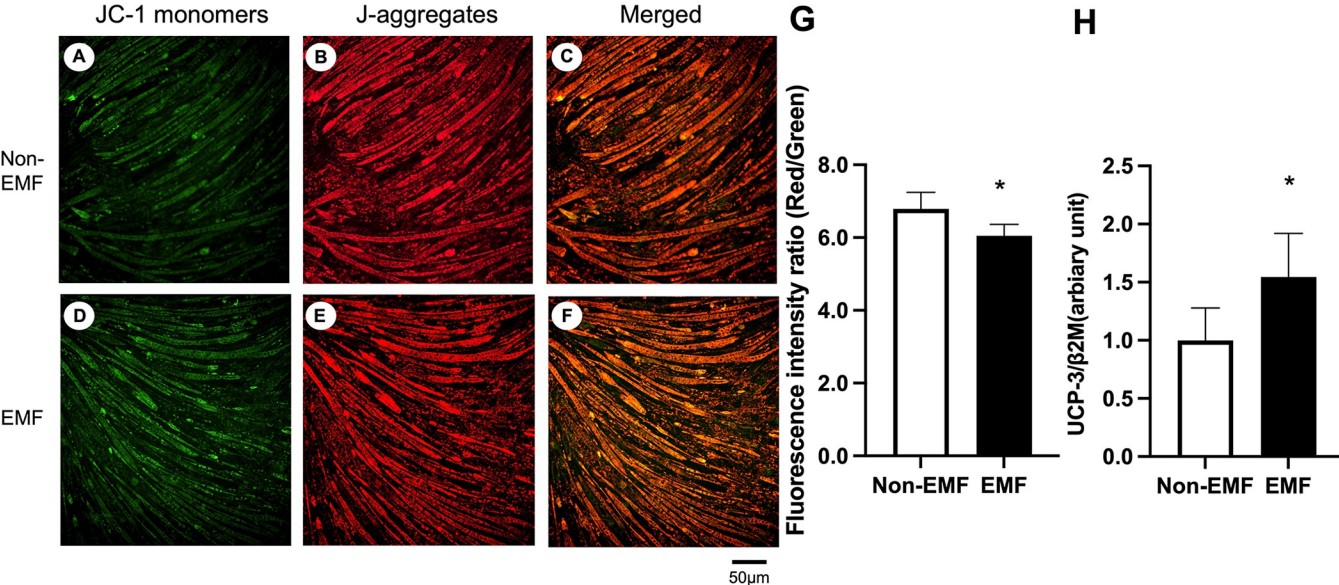

**Fig 7. Mitochondrial membrane potential and UCP-3 mRNA levels in C2C12 Myotubes.** Representative images of JC-1-stained myotubes from C2C12 cells for the Non-EMF (A to C) and EMF (D to F) groups. The images show the expression of JC-1 monomers (green) (A and D), J-aggregates (red) (B and E), and for the images merged (C and F). The histograms show quantification of J-aggregates/JC-1 monomers (G) UCP-3/β-2 microglobulin mRNA expression (H). Values are expressed as mean ± SD. Abbreviations: Non-EMF, C2C12 with no alternating current electromagnetic field exposure; EMF, C2C12 exposed to an alternating current electromagnetic field. * $P < 0.05$ vs. Non-EMF.

increase in UCP-3 expression. Consequently, intramyocellular lipid accumulation was reduced in the muscles of mice fed a high-fat diet, thus attenuating increased insulin resistance and increased circulating glucose levels. These effects are likely to involve increased fatty acid oxidation, mitochondrial metabolism, and mitochondrial uncoupling as a result of AMPKα phosphorylation.

In the present study, alternating current electromagnetic field exposure was shown to be an effective intervention in lessening intramyocellular lipid accumulation, increased insulin resistance, and increased blood glucose levels induced by a high-fat diet in adult mice. These beneficial effects were achieved by increasing fatty acid oxidation, mitochondrial metabolism, and mitochondrial uncoupling following AMPKα phosphorylation. In general, exercise is a practical intervention for increasing AMPKα phosphorylation via increased AMP protein, preventing intramyocellular lipid accumulation [18, 42, 57–59]. Exercise, however, may be difficult for obese patients or hyperglycemic patients with severe peripheral neuropathies or severe micro- and macrovascular complications [17]. Thus, it is important to note that the alternating current electromagnetic field exposure used in the present study changed the AMPKα phosphorylation levels in adult mice under normal resting cage conditions, i.e., without any exercise interventions. For metabolic disorders such as obesity and type 2 diabetes, patients have low motivation to exercise and do not continue to exercise, which is a problem in clinical practice. However, the data indicated that alternating current electromagnetic fields prevented insulin resistance and hyperglycemia without invasive and muscle contraction like an exercise. Furthermore, this alternating current electromagnetic may be an effective therapeutic regimen in rehabilitative strategies for various metabolic disorders because the equipment has an intensity that conforms to the Japanese Industrial Standards (JIS) and can be safely and continuously performed.

A limitation of the present study is that it is unclear whether the effect of mitochondrial metabolic capacity was caused directly by the alternating magnetic field or through muscle-

type shifting. Future studies of alternating magnetic fields on muscle-type shifting should also be performed under these conditions. Another limitation is that we only studied the effects on AMPK phosphorylation and do not know what the effects would be on the AMPK pathway, such as acetyl-CoA carboxylase (ACC). Future studies, however, should be performed to determine the AMPK pathway individually for a detailed mechanism of alternation magnetic fields.

## Conclusion

The present results suggest that alternating current electromagnetic field exposure might be an effective countermeasure to insulin resistance and hyperglycemia by preventing intramyocellular lipid accumulation associated with high-fat feeding. These mechanisms suggest due to the increased transport of fatty acids into the mitochondria, mitochondria metabolism, and mitochondria uncoupling induced by AMPK phosphorylation. Based on these results, further clinical efficacy studies should be conducted to investigate the effects of alternating current electromagnetic field on insulin resistance and hyperglycemia associated with various lipid metabolic diseases, such as type 2 diabetes mellitus.

## Supporting information

**S1 Raw images. Raw images of western blots for Figs 5 and 6.**
(TIFF)

**S1 File.**
(XLSX)

## Author Contributions

**Conceptualization:** Ryosuke Nakanishi, Hidemi Fujino.

**Data curation:** Ryosuke Nakanishi, Masayuki Tanaka, Badur un Nisa, Sayaka Shimizu, Takumi Hirabayashi.

**Formal analysis:** Ryosuke Nakanishi.

**Funding acquisition:** Ryosuke Nakanishi.

**Investigation:** Ryosuke Nakanishi.

**Methodology:** Ryosuke Nakanishi, Hidemi Fujino.

**Project administration:** Hidemi Fujino.

**Software:** Ryosuke Nakanishi.

**Supervision:** Hidemi Fujino.

**Validation:** Minoru Tanaka, Hidemi Fujino.

**Writing – original draft:** Ryosuke Nakanishi.

**Writing – review & editing:** Ryosuke Nakanishi, Masayuki Tanaka, Minoru Tanaka, Noriaki Maeshige, Roland R. Roy, Hidemi Fujino.

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
