## [Decision Letter · Decision Letter 0]

2 May 2023

PONE-D-23-04569Alternating current electromagnetic field exposure lessens intramyocellular lipid accumulation due to high-fat feeding via enhanced lipid metabolism in micePLOS ONE

Dear Dr. Fujino,

Thank you for submitting your manuscript to PLOS ONE. After careful consideration, we feel that it has merit but does not fully meet PLOS ONE’s publication criteria as it currently stands. Therefore, we invite you to submit a revised version of the manuscript that addresses the points raised during the review process.

We look forward to receiving your revised manuscript.

Kind regards,

Kyung-Wan Baek, Ph.D.

Academic Editor

PLOS ONE

Journal Requirements:

"This study was supported by Grants-in-Aid for Scientific Research from the Japanese Ministry of Education, Culture, Sports, Science, and Technology (JSPS KAKENHI, grant number 20K19331, 20KK0227, 22H03453 and 22K19752)."

"This study was supported by Grants-in-Aid for Scientific Research from the Japanese Ministry of Education, Culture, Sports, Science, and Technology (JSPS KAKENHI, grant number 20K19331, 20KK0227, 22H03453 and 22K19752)."

"This study was supported by Grants-in-Aid for Scientific Research from the Japanese Ministry of Education, Culture, Sports, Science, and Technology (JSPS KAKENHI, grant number 20K19331, 20KK0227, 22H03453 and 22K19752)."

Additional Editor Comments:

This manuscript can be published in PLoS One if the reviewer's questions are properly answered and the revision instruction is sufficiently followed. Please fully reflect the opinions of the reviewers so that appropriate revisions can be made. In particular, the opinion of Reviewer #1, who made the Accept decision, should be fully considered.

Reviewers' comments:

Reviewer's Responses to Questions

**Comments to the Author**

1. Is the manuscript technically sound, and do the data support the conclusions?

Reviewer #1: Yes

Reviewer #2: Yes

2. Has the statistical analysis been performed appropriately and rigorously? 

Reviewer #1: Yes

Reviewer #2: Yes

3. Have the authors made all data underlying the findings in their manuscript fully available?

Reviewer #1: Yes

Reviewer #2: Yes

4. Is the manuscript presented in an intelligible fashion and written in standard English?

Reviewer #1: Yes

Reviewer #2: Yes

5. Review Comments to the Author

Reviewer #1: Experiments using Male C57/BL6 mice and C2C12 cells found that alternating current electrical field can reduce intracellular lipids by increasing fat consumption and accumulation. The experimental design was rigorous, using five groups: normal diet (ND), normal diet and exposure to alternating current electromagnetic fields (ND+EMF), high fat diet (HFD) or high fat diet, and alternating current electromagnetic fields (HFD+EMF).

The experiment suggest it should be able to treat insulin resistance and hyperglycemia by exposing to alternating current electrical field. Its mechanism is mainly AMPK α Data analysis methods for phosphorylation and increased UCP-3 protein expression are very rigorous. Although there have been articles describing the treatment of insulin resistance with electromagnetic fields, the mechanism of its research is different from this article.

This paper explained the experiment about mice exposure to alternating current electromagnetic fields with normal diet or high fat diet fluently and get a related conclusion. A lot of test has been done. Like Insulin Tolerance Tests and Oral Glucose Tolerance Tests, Plasma Biochemistry, Histology and Immunohistochemical Analyses ,and Western Blot Analysis. I recommend this article because it has enough workload and the experimental design is reasonable. At the same time, I have a few questions about this paper.

1 How to determine the intensity of electromagnetic fields

2 Why do you need to conduct cell experiments

3. Can you describe the C2C12 cell in detail? I only found out after consulting the information that it is a kind of mouse cell.

4. In medicine, AMPK refers to an AMP activated protein kinase, which is a heterotrimeric protein composed of α、β、γ Consisting of three subunits, widely present throughout the body. Uncoupling protein 3 (UCP3) is a proton carrier on the inner membrane of mitochondria .You could tell more about Professional terms like AMPK α and UCP3. Otherwise the reader has to search for them.

Reviewer #2: Nakanishi et al. studied the effect of alternating current EM field exposure on intramyocellular lipid accumulation in lean and obese insulin resistant mice. They found that in obese insulin resistant mice EMF decreased intramyocellular lipid levels, however in lean mice EMF did not have any significant effect on intramyocellular lipid levels. They also showed that EMF increased the expression of proteins correlated with intracellular lipid metabolism. The manuscript is interesting and has merit, however, it needs additional improvements.

1. Please provide information how strong was the EMF and discuss how such strength would be potentially transferable to humans and clinical practice.

2. Is the size of the field written correctly 328 square mm seem a bit large for a field of view. Please provide details what microscope was used with what objective, what kind of camera, what was the resolution of images. Please briefly describe also how image analysis was performed for each analysis. The methods section is in general not detailed enough to enable reproducibility.

3. For data regarding animals please use means +- SD instead of SEM. Can you also provide exact data from previous study based on which the sample size was calculated. Did you perform p value correction for multiple comparisons each time? Or just when interaction was found?

4. Was the activity of the animals monitored for?

5. In representative figure 3 there are many artefacts on the images. How did you account for these artefacts during analysis?

6. Regarding the analysis of intramyocellular lipid accumulation, SDH activity and FAT/CD36 expression that is based on microscopy, there is a potential very problematic bias. Tibialis anterior muscle is in mice very heterogenous, consisting of portions that are predominantly composed of oxidative and portions that are predominantly composed of more glycolytic fibre types. Therefore, during the fields of view sampling, a significant bias can be made. This can be overcome by systematic random sampling of more fields of view (at least 5), by analysis of whole muscle cross-section or by fibre type specific analysis.

7. Since no fibre type analysis has been performed it is not possible to know whether observed changes are due to effect of EMF on intramyocellular lipid accumulation or fibre type shifting. This is well demonstrated in figure 4 where proportion of fibres with greater SDH activity is seen, however, it is not known whether SDH activity of fibres of same types is increased. This is even more important, because it has been shown that in mice different fibre types in different muscles accumulate lipids differentially during HFD. Therefore, other muscles, for example gluteus maximus muscle could be more appropriate for such analysis since it is less affected by increased weight bearing due to increased mass of the animals.

8. In figure 3 the name of the y axis could be “IMCL content index”.

9. What is the y axis unit in SDH activity in figure 4? Average grey value of 8 bit pixles?

10. It would be beneficial to also confirm whether downstream targets of AMPK such as ACC were also affected.

11. Please state also limitations of your study.

12. There is no data availability statement.

13. The conclusion " These mechanisms are most likely due to the increased transport of fatty acids into the mitochondria, mitochondria metabolism, and mitochondria uncoupling induced by AMPK phosphorylation." should be a bit ameliorated since the results of these study only suggest these, and are not sufficiently rigorous.

6. PLOS authors have the option to publish the peer review history of their article (what does this mean?). If published, this will include your full peer review and any attached files.

Reviewer #1: No

Reviewer #2: No

---

## [Author Response · Author response to Decision Letter 0]

26 May 2023

【Response to the Academic Editor 's comments】

#1. 1. Please ensure that your manuscript meets PLOS ONE's style requirements, including those for file naming. The PLOS ONE style templates can be found at 

Response: We are sorry about incorrect format. We have added the format such as position of Table, Figure and deleted the Acknowledgement sentence.

#2. Please note that PLOS ONE has specific guidelines on code sharing for submissions in which author-generated code underpins the findings in the manuscript. In these cases, all author-generated code must be made available without restrictions upon publication of the work. Please review our guidelines at https://journals.plos.org/plosone/s/materials-and-software-sharing#loc-sharing-code and ensure that your code is shared in a way that follows best practice and facilitates reproducibility and reuse.

Response: 

We have added the RRID for each antibody. Please check the text. 

#3. Thank you for stating the following financial disclosure: 

"This study was supported by Grants-in-Aid for Scientific Research from the Japanese Ministry of Education, Culture, Sports, Science, and Technology (JSPS KAKENHI, grant number 20K19331, 20KK0227, 22H03453 and 22K19752)."

Response: 

We have stated the role the funders took in the study to the cover letter.

[The funder and role of this study]

Ryosuke Nakanishi (20K19331) is involved in all processes involved in this experiment (conceived, designed, performed, analyzed, interpreted results, prepared figures, drafted manuscript, edited and revised manuscript).

Hidemi Fujino (20KK0227, 22H03453 and 22K19752) contributed to conceiving and designing this experiment and correcting the manuscript. In addition, the author supervised all of this experiment as the corresponding author.

#4. Thank you for stating the following in the Acknowledgments Section of your manuscript: 

"This study was supported by Grants-in-Aid for Scientific Research from the Japanese Ministry of Education, Culture, Sports, Science, and Technology (JSPS KAKENHI, grant number 20K19331, 20KK0227, 22H03453 and 22K19752)."

"This study was supported by Grants-in-Aid for Scientific Research from the Japanese Ministry of Education, Culture, Sports, Science, and Technology (JSPS KAKENHI, grant number 20K19331, 20KK0227, 22H03453 and 22K19752)."

Response: We have deleted the funding Statement in the acknowledgment section according to your point out. We have Added the following text to our cover letter. We appreciate for your correction. 

“This study was supported by Grants-in-Aid for Scientific Research from the Japanese Ministry of Education, Culture, Sports, Science, and Technology (JSPS KAKENHI, grant number 20K19331, 20KK0227, 22H03453 and 22K19752).”

#5. In your Data Availability statement, you have not specified where the minimal data set underlying the results described in your manuscript can be found. PLOS defines a study's minimal data set as the underlying data used to reach the conclusions drawn in the manuscript and any additional data required to replicate the reported study findings in their entirety. All PLOS journals require that the minimal data set be made fully available. For more information about our data policy, please see http://journals.plos.org/plosone/s/data-availability.

Response: Thank you for your advice. We now provide the supplement data and raw image.

#6. PLOS ONE now requires that authors provide the original uncropped and unadjusted images underlying all blot or gel results reported in a submission’s figures or Supporting Information files. This policy and the journal’s other requirements for blot/gel reporting and figure preparation are described in detail at https://journals.plos.org/plosone/s/figures#loc-blot-and-gel-reporting-requirements and https://journals.plos.org/plosone/s/figures#loc-preparing-figures-from-image-files. When you submit your revised manuscript, please ensure that your figures adhere fully to these guidelines and provide the original underlying images for all blot or gel data reported in your submission. See the following link for instructions on providing the original image data: https://journals.plos.org/plosone/s/figures#loc-original-images-for-blots-and-gels. 

Response: We now uploaded the raw image data.

 

REVIEWER COMMENTS:

Response to the comments of reviewer #1.

We thank the Referee for the insightful comments on our paper. The comments have helped us to significantly improve the manuscript.

Revisions

#1. How to determine the intensity of electromagnetic fields.

Response: The clinical practice uses high-intensity magnetic fields as medical devices (1.5-2.0 T are used to treat depression and Parkinson's disease (1), 272 mT stimulation was used to treat stress urinary incontinence (2), 3.0 T stimulation was used to treat female urinary incontinence (3)). The device used in this research has a lower strength than these medical devices. The Japanese Industrial Standards (JIS) stipulate that the maximum magnetic flux density should be 180mT or less to ensure human user safety. Therefore, by setting it to 180mT or less, it is a strength that can be used safely not only in clinical practice but also for health promotion. 

References

1. Rossi, S., Hallett, M., Rossini, P. M., Pascual-Leone, A. and Group, S. o. T. C. (2009) Safety, ethical considerations, and application guidelines for the use of transcranial magnetic stimulation in clinical practice and research. Clinical neurophysiology. 120; 2008-2039.

2. Yamanishi, T., Suzuki, T., Sato, R., Kaga, K., Kaga, M. and Fuse, M. (2019) Effects of magnetic stimulation on urodynamic stress incontinence refractory to pelvic floor muscle training in a randomized sham‐controlled study. LUTS: Lower Urinary Tract Symptoms. 11; 61-65.

3. Braga, A., Castronovo, F., Caccia, G., Papadia, A., Regusci, L., Torella, M., et al. (2022) Efficacy of 3 Tesla Functional Magnetic Stimulation for the Treatment of Female Urinary Incontinence. Journal of Clinical Medicine. 11; 2805.

#2. Why do you need to conduct cell experiments

Response: We confirmed that the expression of UCP3, an uncoupling protein, was increased in mice experiments. However, it is necessary to confirm whether uncoupling in mitochondria actually occurs in living cells immediately after irradiation. Therefore, we performed cell experiments and proved that uncoupling occurred.

#3. Can you describe the C2C12 cell in detail? I only found out after consulting the information that it is a kind of mouse cell.

Response: C2C12 cells published by DAVID YAFFE & ORA SAXEL in 1977 (1) and are isolated from the muscle of mice, and are able to differentiate into skeletal muscle cells. C2C12 cells have often been used to study the mitochondrial uncoupling of muscle in vitro (2-4).

1. Yaffe, D. and Saxel, O. (1977) Serial passaging and differentiation of myogenic cells isolated from dystrophic mouse muscle. Nature. 270; 725-727.

2. Minners, J., Lacerda, L., McCarthy, J., Meiring, J. J., Yellon, D. M. and Sack, M. N. (2001) Ischemic and pharmacological preconditioning in Girardi cells and C2C12 myotubes induce mitochondrial uncoupling. Circulation research. 89; 787-792.

3. Fan, X., Hussien, R. and Brooks, G. A. (2010) H2O2-induced mitochondrial fragmentation in C2C12 myocytes. Free Radical Biology and Medicine. 49; 1646-1654.

4. Wong, H. S., Chen, J., Leong, P. K., Leung, H. Y., Chan, W. M. and Ko, K. M. (2015) A Cistanches Herba fraction/β-Sitosterol causes a redox-sensitive induction of mitochondrial uncoupling and activation of adenosine monophosphate-dependent protein kinase/peroxisome proliferator-activated receptor γ coactivator-1 in C2C12 myotubes: A possible mechanism underlying the weight reduction effect. Evidence-Based Complementary and Alternative Medicine. 2015.

#4. In medicine, AMPK refers to an AMP activated protein kinase, which is a heterotrimeric protein composed of α、β、γ Consisting of three subunits, widely present throughout the body. Uncoupling protein 3 (UCP3) is a proton carrier on the inner membrane of mitochondria .You could tell more about Professional terms like AMPK α and UCP3. Otherwise the reader has to search for them.

Response:

We appreciate your kind advice. Per your advice, we have added the sentence about the detailed explanation of AMPK and UCP3. In addition, we have added the reference relates to these sentences.

Page 5, Lines 59-60: “Uncoupling proteins (UCPs) constitute a mitochondrial carrier proteins subgroup (UCP-1–UCP-5) located in the inner mitochondrial membrane[13].”

Pages 31, Lines 418-421: “These factors have been shown to interact with and decrease phosphorylated AMP-activated protein kinase (AMPK) α in the muscles of mice in response to long-term feeding with a high-fat diet, consistent with the present results”

Page 32, Lines 428-431: “AMPK is a heterotrimeric serine-threonine kinase composed of the catalytic α- and noncatalytic β- and γ-subunits [51]. AMPK plays a role in maintaining energy homeostasis and is a target for treating various metabolic disorders [52]. Thus, AMPK activation likely to mitigates metabolic impairments by controlling energy homeostasis.”

Reference

・[13] Vidal-Puig, A., Solanes, G., Grujic, D., Flier, J. S. and Lowell, B. B. (1997) UCP3: an uncoupling protein homologue expressed preferentially and abundantly in skeletal muscle and brown adipose tissue. Biochemical and biophysical research communications. 235; 79-82.

・[51] Hardie, D. G., Ross, F. A. and Hawley, S. A. (2012) AMPK: a nutrient and energy sensor that maintains energy homeostasis. Nature reviews Molecular cell biology. 13; 251-262.

・[52] Fogarty, S. and Hardie, D. (2010) Development of protein kinase activators: AMPK as a target in metabolic disorders and cancer. Biochimica et biophysica acta (bba)-proteins and proteomics. 1804; 581-591.

 

Response to the comments of reviewer #2

We thank the Referee for carefully reading our manuscript and for the helpful comments. 

#1. Please provide information how strong was the EMF and discuss how such strength would be potentially transferable to humans and clinical practice.

Response: The clinical practice uses high-intensity magnetic fields as medical devices (1.5-2.0 T are used to treat depression and Parkinson's disease (1), 272 mT stimulation was used to treat stress urinary incontinence (2), 3.0 T stimulation was used to treat female urinary incontinence (3)). The device used in this research has a lower strength than these medical devices. The Japanese Industrial Standards (JIS) stipulate that the maximum magnetic flux density should be 180mT or less to ensure human user safety. Therefore, by setting it to 180mT or less, it is a strength that can be used safely not only in clinical practice but also for health promotion. 

Furthermore, this intensity can be exposed without feeling fatigued or stressed. Obesity and type 2 DM patients have low motivation to exercise and do not continue to exercise, which is a problem in clinical practice. We believe that if these patients can be treated by being exposed to magnetic fields, they can continue treatment regardless of their motivation for exercise anywhere. We have added the sentence about potentially transferable to humans and clinical practice in the discussion.

Page 16, Lines 472-480: “For metabolic disorders such as obesity and type 2 diabetes, patients have low motivation to exercise and do not continue to exercise, which is a problem in clinical practice. However, the data indicated that alternating current electromagnetic fields prevented insulin resistance and hyperglycemia without invasive and muscle contraction like an exercise. Furthermore, this alternating current electromagnetic may be an effective therapeutic regimen in rehabilitative strategies for various metabolic disorders because the equipment has an intensity that conforms to the Japanese Industrial Standards (JIS) and can be safely and continuously performed.” 

references

4. Rossi, S., Hallett, M., Rossini, P. M., Pascual-Leone, A. and Group, S. o. T. C. (2009) Safety, ethical considerations, and application guidelines for the use of transcranial magnetic stimulation in clinical practice and research. Clinical neurophysiology. 120; 2008-2039.

5. Yamanishi, T., Suzuki, T., Sato, R., Kaga, K., Kaga, M. and Fuse, M. (2019) Effects of magnetic stimulation on urodynamic stress incontinence refractory to pelvic floor muscle training in a randomized sham‐controlled study. LUTS: Lower Urinary Tract Symptoms. 11; 61-65.

6. Braga, A., Castronovo, F., Caccia, G., Papadia, A., Regusci, L., Torella, M., et al. (2022) Efficacy of 3 Tesla Functional Magnetic Stimulation for the Treatment of Female Urinary Incontinence. Journal of Clinical Medicine. 11; 2805.

#2. Is the size of the field written correctly 328 square mm seem a bit large for a field of view. Please provide details what microscope was used with what objective, what kind of camera, what was the resolution of images. Please briefly describe also how image analysis was performed for each analysis. The methods section is in general not detailed enough to enable reproducibility.

Response: We are sorry that the following sentence in the initial submission was a mistake: “The lipid area was calculated randomly within the section for each of 3 different fields (328mm2/ field), and a mean percentage was calculated for each muscle”. In fact, muscle fibers were analyzed in 94.5mm2 (340.4um(wide)×272.3um(height) /field for SDH stain and oil red O stain measurement. However, a value of 94.5 mm2 was used in the actual calculations. The section was viewed and captured with a CX41 microscope (Olympus, Tokyo, Japan), and set the ocular lens: x10, objective lens: x20. Furthermore, we have export images as 600 dpi TIFF files using the software after the calculation.

We have now modified the sentence as follows:

Page 12, lines 164-167:” The sections were captured with a microscope (CX41; Olympus, Tokyo, Japan, objective lens: x20), and were quantified using Image J software (NIH, Bethesda, MD). Raw images were exported as 600 dpi TIFF files using software after quantified.”

In addition, muscle fibers were analyzed in 211.6 mm2 (581.81um(wide)×363.63um(height) /field for immunohistochemistry stain measurement. The section was viewed and captured with a BX51 microscope (Olympus, Tokyo, Japan), and set the ocular lens: x10, objective lens: x40. Furthermore, we have exported images as 600 dpi TIFF files using the software after the calculation. We have also added the following sentence to clarify the analysis method.

Page 14, Lines 181-184: “Immunohistochemical analyses were conducted as previously described [29] The sections were captured with a fluorescence microscope (BX51; Olympus) and quantified by Image J software. Raw images were exported as 600 dpi TIFF files using software after quantified.”

Page 14, Lines 191-194: “The cell surface immunofluorescence was captured randomly within the section for each of the five different fields (211.6mm2/ field), and a mean percentage was calculated for each muscle (total area occupied by immunofluorescence of a muscle fiber × 100/total cross-sectional area of the fiber) [30].”

#3. For data regarding animals please use means +- SD instead of SEM. Can you also provide exact data from previous study based on which the sample size was calculated. Did you perform p value correction for multiple comparisons each time? Or just when interaction was found?

Response:

[About SD] We now correct all results as means ± SD (not SEM) and have modified several sentences, Table, and Figures. 

[About sample size calculation] We have corrected the citation of the article and analysis. The previous study reported that the partial η2 of the impact of diet in experiments in which mice aged 4 to 20 weeks were given HFD was 0.33 i.e., effect size f = 0.70. α＝0.05, power = 0.80, Numerator df = 1, Number of groups = 4. These results indicated total sample size =19. 

[about p-value] We performed p-value correction for multiple comparisons each time. The table are five data sets with no significant difference in interactions.

 Normal diet High-fat diet 

 Non EMF EMF Non EMF EMF Interaction p-value of 

HFD V.S. HFD+EMF

Calorie intake (Kcal/day) 8.9 ± 0.2 8.9 ± 0.2 12.7 ± 0.3* 12.3 ± 0.2* >0.9999 >0.9999

Body mass (g) 24.2 ± 0.2 24.3 ± 0.4 39.3 ± 1.8* 36.5 ± 0.7* 0.2 0.5

Fat mass (mg) 356.8 ± 7.1 402.0 ± 21.3 2907.0 ± 185.6* 2380.8 ± 131.4* 0.07 0.1

Absolute tibial anterior 

muscle mass (mg) 42.3 ± 1.3 44.0 ± 0.9 48.2 ± 0.5 44.6 ± 0.7 0.03 0.1

Relative tibial anterior 

muscle mass (mg/g) 1.8 ± 0.05 1.8 ± 0.04 1.2 ± 0.06* 1.2 ± 0.03* 0.9 >0.9999

#4. Was the activity of the animals monitored for?

Response: We monitored the activity of mice every day, and no change in activity compared with non-EMF. In addition, mice were not observed to lose appetite or body weight. However, we do not have objective data such as a laser activity meter.

#5. In representative figure 3 there are many artefacts on the images. How did you account for these artefacts during analysis?

Response: If the artifact is severe and difficult to count, we exclude it from the analysis.

#6. Regarding the analysis of intramyocellular lipid accumulation, SDH activity and FAT/CD36 expression that is based on microscopy, there is a potential very problematic bias. Tibialis anterior muscle is in mice very heterogenous, consisting of portions that are predominantly composed of oxidative and portions that are predominantly composed of more glycolytic fibre types. Therefore, during the fields of view sampling, a significant bias can be made. This can be overcome by systematic random sampling of more fields of view (at least 5), by analysis of whole muscle cross-section or by fibre type specific analysis.

Response: Thank you for your kind guidance. According to your advice, I increased the field of view to analyze from three fields to five fields. The overall values changed, but the statistical differences remained the same. Please check Figure 3-5.

#7. Since no fibre type analysis has been performed it is not possible to know whether observed changes are due to effect of EMF on intramyocellular lipid accumulation or fibre type shifting. This is well demonstrated in figure 4 where proportion of fibres with greater SDH activity is seen, however, it is not known whether SDH activity of fibres of same types is increased. This is even more important, because it has been shown that in mice different fibre types in different muscles accumulate lipids differentially during HFD. Therefore, other muscles, for example gluteus maximus muscle could be more appropriate for such analysis since it is less affected by increased weight bearing due to increased mass of the animals.

Response: As you pointed out, it is not clear whether the direct effect of the alternating magnetic field increased the mitochondrial metabolic capacity, or whether the metabolic capacity increased due to the fiber type shifting.

In fact, previous research has shown that EDL muscle typeIIA increases by promoting AMPK with AICAR(1). These results indicated that the alternating magnetic field is possible to increase metabolic capacity due to the fiber type shifting by AMPK activation. Further research is needed to draw these conclusions. This content is described in the discussion as a limitation of the research. Thanks for the great point out．

1. Ljubicic, V., Miura, P., Burt, M., Boudreault, L., Khogali, S., Lunde, J. A., et al. (2011) Chronic AMPK activation evokes the slow, oxidative myogenic program and triggers beneficial adaptations in mdx mouse skeletal muscle. Human molecular genetics. 20; 3478-3493.

Page 35, lines 481-484:” A limitation of the present study is that it is unclear whether the effect of mitochondrial metabolic capacity was caused directly by the alternating magnetic field or through muscle-type shifting. Future studies of alternating magnetic fields on muscle-type shifting should also be performed under these conditions.”

#8. In figure 3 the name of the y axis could be “IMCL content index”.

Response: According to the point out, we have corrected the in figure 3 the name of the y from “IMCL” to “IMCL content index”

#9. What is the y axis unit in SDH activity in figure 4? Average grey value of 8 bit pixles?.

Response: We are sorry about missing the y-unit in Figure 4. Yes, we analyzed using the Average grey value of 8-bit pixels. Previous studies express using various labels below.

We have corrected the y-axis title to “SDH activity (O.D.)”. In addition, we have added the sentence preventing the misreading for the reader.

Page 13, lines 178-180; The SDH activity was captured randomly within the section for each of the five different fields (92.5mm2/ field), and was converted to 8-bit grayscale and was quantified as a mean gray value.

1. mean gray values (A.U.)

Giacomello, E., Crea, E., Torelli, L., Bergamo, A., Reggiani, C., Sava, G., et al. (2020) Age dependent modification of the metabolic profile of the tibialis anterior muscle fibers in C57BL/6J mice. International journal of molecular sciences. 21; 3923. 

1. SDH (relative unit)

Delgadillo-Puga, C., Sánchez-Castillo, D. R., Cariño-Cervantes, Y. Y., Torre-Villalvazo, I., Tovar-Palacio, C., Vásquez-Reyes, S., et al. (2023) Vachellia farnesiana Pods or a Polyphenolic Extract Derived from Them Exert Immunomodulatory, Metabolic, Renoprotective, and Prebiotic Effects in Mice Fed a High-Fat Diet. International journal of molecular sciences. 24; 7984.

2. SDH activity (O.D.)

Eshima, H., Tamura, Y., Kakehi, S., Kakigi, R., Kawamori, R. and Watada, H. (2021) Maintenance of contractile force and increased fatigue resistance in slow-twitch skeletal muscle of mice fed a high-fat diet. Journal of Applied Physiology. 130; 528-536.

3. SDH activity (OD)

Kanazashi, M., Tanaka, M., Nakanishi, R., Maeshige, N. and Fujino, H. (2019) Effects of astaxanthin supplementation and electrical stimulation on muscle atrophy and decreased oxidative capacity in soleus muscle during hindlimb unloading in rats. The journal of physiological sciences: JPS. 69; 757-767.

#10. It would be beneficial to also confirm whether downstream targets of AMPK such as ACC were also affected.

Response: We appreciate your kind opinion, but there are no samples left, and analysis is not possible. According to your point out, ACC1 or ACC2 has a critical role in fatty acid metabolism (ACC1 regulates fatty acid synthesis, and ACC2 regulates only fatty acid oxidation) (1). A detailed mechanism by alternating magnetic fields can be obtained by clarifying the AMPK pathway in further research in the future. These results indicated that We had added the sentence as the limitation sentence instead.

Pages 35-36 lines 484-487: Another limitation is that we only studied the effects on AMPK phosphorylation and do not know what the effects would be on the AMPK pathway, such as acetyl-CoA carboxylase (ACC). Future studies, however, should be performed to determine the AMPK pathway individually for a detailed mechanism of alternation magnetic fields.

1. Steinberg, G. R. and Schertzer, J. D. (2014) AMPK promotes macrophage fatty acid oxidative metabolism to mitigate inflammation: implications for diabetes and cardiovascular disease. Immunology and cell biology. 92; 340-345.

2. 

#11. Please state also limitations of your study.

Response: Based on referee’s comments, we have created the following "limitations of this study".

Pages 35-36, Lines 481-488: A limitation of the present study is that it is unclear whether the effect of mitochondrial metabolic capacity was caused directly by the alternating magnetic field or through muscle-type shifting. Future studies of alternating magnetic fields on muscle-type shifting should also be performed under these conditions. Another limitation is that we only studied the effects on AMPK phosphorylation and do not know what the effects would be on the AMPK pathway, such as acetyl-CoA carboxylase (ACC). Future studies, however, should be performed to determine the AMPK pathway individually for a detailed mechanism of alternation magnetic fields.

#12. There is no data availability statement.

We uploaded the “Supporting Information files” to the platform.

# 13. The conclusion " These mechanisms are most likely due to the increased transport of fatty acids into the mitochondria, mitochondria metabolism, and mitochondria uncoupling induced by AMPK phosphorylation." should be a bit ameliorated since the results of these study only suggest these, and are not sufficiently rigorous.

Response: According to your advice, we have corrected the words from “ are most likely” to “suggest”.

Page 36, lines 495-497: These mechanisms suggest due to the increased transport of fatty acids into the mitochondria, mitochondria metabolism, and mitochondria uncoupling induced by AMPK phosphorylation.

---

## [Decision Letter · Decision Letter 1]

11 Jul 2023

Alternating current electromagnetic field exposure lessens intramyocellular lipid accumulation due to high-fat feeding via enhanced lipid metabolism in mice

PONE-D-23-04569R1

Dear Dr. Fujino,

We’re pleased to inform you that your manuscript has been judged scientifically suitable for publication and will be formally accepted for publication once it meets all outstanding technical requirements.

Kind regards,

Kyung-Wan Baek, Ph.D.

Academic Editor

PLOS ONE

Additional Editor Comments (optional): I am genuinely pleased to inform you that your manuscript has been judged suitable for publication in PLoS One. Comments from additional reviewers may be disregarded, but please refer to them if necessary.

The revised manuscript was judged suitable for publication in PLoS One.

Comments from additional reviewers may be disregarded, but please refer to them if necessary.

Reviewers' comments:

Reviewer's Responses to Questions

**Comments to the Author**

1. If the authors have adequately addressed your comments raised in a previous round of review and you feel that this manuscript is now acceptable for publication, you may indicate that here to bypass the “Comments to the Author” section, enter your conflict of interest statement in the “Confidential to Editor” section, and submit your "Accept" recommendation.

Reviewer #2: All comments have been addressed

Reviewer #3: All comments have been addressed

2. Is the manuscript technically sound, and do the data support the conclusions?

Reviewer #2: Yes

Reviewer #3: Yes

3. Has the statistical analysis been performed appropriately and rigorously? 

Reviewer #2: Yes

Reviewer #3: Yes

4. Have the authors made all data underlying the findings in their manuscript fully available?

Reviewer #2: Yes

Reviewer #3: Yes

5. Is the manuscript presented in an intelligible fashion and written in standard English?

Reviewer #2: Yes

Reviewer #3: Yes

6. Review Comments to the Author

Reviewer #2: (No Response)

Reviewer #3: The opinions of the editor and two reviewers were appropriately reflected. Please finally check minor typo errors or figure resolution. I expect this paper to be able to publish on PLoS One.

7. PLOS authors have the option to publish the peer review history of their article (what does this mean?). If published, this will include your full peer review and any attached files.

Reviewer #2: **Yes: **Nejc Umek

Reviewer #3: **Yes: **Jeong-An Gim

---

## [Editor Report · Acceptance letter]

19 Jul 2023

PONE-D-23-04569R1 

Alternating current electromagnetic field exposure lessens intramyocellular lipid accumulation due to high-fat feeding via enhanced lipid metabolism in mice 

Dear Dr. Fujino:

I'm pleased to inform you that your manuscript has been deemed suitable for publication in PLOS ONE. Congratulations! Your manuscript is now with our production department. 

Kind regards, 

on behalf of

Dr. Kyung-Wan Baek 

Academic Editor

PLOS ONE